# The Deployment of a Newly Developed Proximal Release-Type Colonic Stent Is Feasible for Malignant Colorectal Obstruction near the Anal Verge: A Single-Center Preliminary Study

**DOI:** 10.3390/jcm11061675

**Published:** 2022-03-17

**Authors:** Kaoru Wada, Toshio Kuwai, Syuhei Sugata, Takuro Hamada, Riho Moriuchi, Yuzuru Tamaru, Ryusaku Kusunoki, Atsushi Yamaguchi, Hirotaka Kouno, Sauid Ishaq, Hiroshi Kohno

**Affiliations:** 1Department of Gastroenterology, National Hospital Organization Kure Medical Center and Chugoku Cancer Center, 3-1 Aoyama-Cho, Kure, Hiroshima 737-0023, Japan; wada.kaoru.qd@mail.hosp.go.jp (K.W.); sugata.shuhei.dy@mail.hosp.go.jp (S.S.); hamada.takuro.dw@mail.hosp.go.jp (T.H.); ishibashi.riho.sn@mail.hosp.go.jp (R.M.); tamaru.yuzuru.vh@mail.hosp.go.jp (Y.T.); kusunoki.ryusaku.mg@mail.hosp.go.jp (R.K.); yamaguchi.atsushi.uc@mail.hosp.go.jp (A.Y.); kouno.hirotaka.pz@mail.hosp.go.jp (H.K.); kouno.hiroshi.tu@mail.hosp.go.jp (H.K.); 2Gastroenterology Department, Dudley Group Hospitals, Birmingham City University, Birmingham B18 7QH, UK; sauid.ishaq@nhs.net

**Keywords:** SEMS, newly developed proximal release-type colonic stent, malignant colorectal obstruction, close to the anal verge

## Abstract

Introduction: Colonic self-expandable metallic stents are widely used to treat malignant colorectal obstructions. Stent placement in lesions near the dentate line causes problems, including severe pain due to difficulty in positioning the stent accurately. Therefore, a proximal release-type stent was developed to overcome this issue, and this preliminary study aimed to investigate its efficacy and safety. Patients and Methods: This research enrolled eight patients with malignant colorectal obstructions up to 10 cm from the anal verge who required placement of the newly developed proximal release-type colonic stent. The primary outcome was the clinical success rate, and the secondary outcomes were the technical success and adverse events rates. Results: The technical and clinical success rates were 87.5% each, and the mean procedure time was 25.5 ± 22.0 min. The mean procedure time in the rectosigmoid colon was significantly longer than that in the rectum. Only one (12.5%) patient had stent migration, and neither anal pain nor tenesmus was observed. Discussion: The stent was highly effective in treating lesions near the anal verge, and it might contribute to the expansion of indications for colorectal stents for lesions near the dentate line. However, the indications for rectosigmoid colon lesions should be cautiously considered.

## 1. Introduction

Colorectal cancer (CRC) is the most common type of cancer worldwide [1], and approximately 8–13% of patients with CRC exhibit symptoms of acute colonic obstruction at the time of diagnosis [2,3,4]. The placement of a self-expandable metallic stent (SEMS) for a malignant colorectal obstruction has been covered by the National Health Insurance in Japan, since January 2012. Moreover, this procedure is widely performed for preoperative decompression (bridge to surgery, BTS) and palliation (PAL) [5,6,7,8,9].

However, there is a risk of severe pain due to proximity to the dentate line because it is technically difficult to deploy the stent precisely with the anal side of the stent exactly aligned with the anal side of the lesion for a malignant colorectal obstruction close to the anal verge. Hence, the placement of the conventional distal release-type SEMS for tumors near the anal verge (particularly <5 cm) is challenging to perform [10,11,12,13,14]. Further, the European Society of Gastrointestinal Endoscopy (ESGE) guidelines state that the success rate of colorectal stenting is low among patients with tumors near the anal verge (<5 cm) due to tenesmus, and the indications should be cautiously considered [15].

To overcome this issue, a new proximal release-type colonic stent that could facilitate adjusting the position of the proximal edge was developed [16]. However, no study to date has evaluated the newly developed proximal release-type colonic stent; thus, its efficacy and safety remain unclear. Therefore, the current study aimed to investigate the efficacy and safety of this colonic stent.

## 2. Patients and Methods

### 2.1. Study Design and Participants

This single-center retrospective observational study was conducted to assess the efficacy and safety of the newly developed proximal release-type colonic stent. Among a total of 60 patients who underwent emergency endoscopy for a malignant colorectal obstruction in our hospital between May 2019 and July 2021, eight patients with malignant colorectal obstructions up to 10 cm from the anal verge who underwent SEMS placement using the proximal release-type colonic stent were included in the analysis. The treatment intent (BTS or PAL) was determined based on the disease stage, comorbidities, age, and patient preference. The scheduled elective resection of primary tumors within 1 year of the stent placement was classified as BTS, whereas those who underwent the stent placement for symptomatic relief without a scheduled surgery were classified as PAL. The time interval between stenting and surgery was dictated primarily by the general condition of the patient and the evaluation of the oral side bowel, which was approximately 2 weeks. Neoadjuvant treatment was considered only in cases with a high risk of invasion.

This study was performed in accordance with the principles of the Declaration of Helsinki in compliance with good clinical practice and local regulations. The study was approved by the Institutional Review Board of the National Hospital Organization Kure Medical Center and Chugoku Cancer Center Institutional. The nature of the procedure was explained to the patients, and informed consent for the procedure and data collection was obtained.

### 2.2. Inclusion and Exclusion Criteria

The registry included patients with a colorectal obstruction within 10 cm of the anal verge caused by CRC or extracolonic cancer that required decompression for BTS or PAL. The diagnosis was based on abdominal radiography, a computed tomography scan, or a colonoscopy. The decision for stent placement or stoma creation was made by each patient. In a shared decision-making process, the risk of stent-related perforation, recurrence rate, overall survival and postoperative mortality, overall complication and permanent stoma rates, rate of laparoscopic one-stage surgery procedures, and technical and clinical failure rates of stenting were considered [15]. The exclusion criteria were enteral ischemia, suspected or impending perforation, an intra-abdominal abscess, any contraindication to endoscopic treatment, and the use of the stent for which it was not indicated.

### 2.3. SEMS Placement Procedures

In the current study, Niti-S (an enteral colonic uncovered stent; Taewoong Medical Co., Gimpo, South Korea), which is a flexible, fine mesh tubular prosthesis made of nitinol wire that contained radiopaque markers on each end and at the center, was used as the proximal release-type colonic stent. The diameter and length of the newly proximal release-type colonic stent were 22 and 70 mm, respectively, and the delivery system was 16 Fr (Figure 1). The procedures were performed in a fluoroscopy room by a certified endoscopist with experience in colonic stenting using the over-the-wire method [17,18]. The stricture length was measured via fluoroscopy using gastrografin as the contrast agent (Figure 2A).

First, the nasal endoscope GIF-XP290N (Olympus Optical Co., Tokyo, Japan) was advanced via the stenosis to the oral side, and a guidewire was placed at the oral side across the stenosis site via the scope (Figure 2B). Next, the scope was removed while leaving the wire in place and reintroduced beside the guidewire (Figure 2C,G). The delivery system was advanced to the appropriate position over the wire under nasal endoscopic and fluoroscopic views (Figure 2D). The stent was released from the proximal side at the point where it was perfectly aligned with the anal side of the tumor and deployed at the precise position (Figure 2E,H). Finally, we assessed the stent placement position via fluoroscopy using a contrast agent. In addition, abdominal radiography was conducted at 24 and 48 h after the procedure to assess for stent migration and poor or failed expansion.

### 2.4. Outcomes and Definitions

The primary outcome was the rate of clinical success, defined as the resolution of the symptoms and radiological findings within 24 h. The ColoRectal Obstruction Scoring System (CROSS) [19] was used to assess the oral intake level and abdominal symptoms before the procedure. The CROSS score is based on the oral intake ability and abdominal symptoms: (a) the requirement of a continuous decompressive procedure, 0; (b) no oral intake, 1; (c) a liquid or an enteral nutrient, 2; (d) soft solids, low residue, 3; and (e) a full diet without symptoms of stricture, 4.

The secondary outcomes were the technical success and adverse events rates. The technical success was defined as an accurate SEMS placement, conferring adequate stricture coverage on the first attempt, free of the procedure-related adverse events. The SEMS placement-related adverse events were perforation, re-obstruction, stent migration, infection, fever, abdominal pain, and tenesmus. Perforation was diagnosed based on clinical, radiological, or intraoperative findings. The early adverse events were defined as adverse events occurring within 7 days, including the day of stenting, and the late adverse events, as those occurring after 7 days. Data on the procedure times, complications, and length of hospitalization after surgery were recorded.

### 2.5. Statistical Analysis

Data about the baseline characteristics and the clinical, tumor, and surgery-related characteristics were expressed as a mean (standard deviation, SD), a median (range), or percentages. Continuous variables were presented as medians and a range or an interquartile range, as appropriate. Continuous and nominal variables were compared using the Wilcoxon signed-rank test. A *p* value of < 0.05 was considered statistically significant. All statistical analyses were performed using JMP16.1.0 (SAS Institute, Inc., Cary, NC, USA).

## 3. Results

### 3.1. Baseline Characteristics of the Study Cohort

Eight consecutive patients were analyzed in this study (Table 1). Among them, three (37.5%) were men and five (62.5%) were women. The median age was 73.5 (range: 64–80) years. Colorectal obstruction was caused by primary colorectal cancer in five patients (including one with anastomotic recurrence), gastric cancer in two, and pancreatic cancer in one. The CROSS scores of 1, 4, 1, and 2 patients were 0, 1, 2, and 3, respectively. The location of the obstruction was the anastomosis of the sigmoid colon in one (12.5%) patient, the sigmoid colon in four (50.0%) patients, the Ra in one (12.5%) patient, and the Rb in two (25.0%) patients. The treatment intent was BTS in four patients and PAL in four patients. The median stricture length was 4 (range: 3.0–10) cm, and the median distance from the dentate line was 5.5 (range: 2.0–9.0) cm.

### 3.2. Short-Term Outcome of SEMS Placement

The technical success rate was achieved in 87.5% of patients (7/8) (Table 2). In one patient, technical failure occurred because of stent migration in the rectosigmoid colon during the procedure, and it was treated with the re-SEMS placement on the same day. The mean procedure time was 25.5 ± 22.0 min. The clinical success rate was 87.5%. One patient, who experienced technical failure, also experienced clinical failure, and stent migration was observed during the elective surgery. Regarding other early adverse events correlated with the SEMS placement, neither anal pain nor tenesmus occurred, and there were no adverse effects associated with elective surgery in the BTS cases. The mean CROSS score was 1.5 before the SEMS placement and 4.0 after the SEMS placement. Seven (87.5%) patients could tolerate oral intake from the third day after the SEMS placement. However, one patient could not do so due to worsening of the primary advanced gastric cancer.

In the subgroup analysis according to the tumor location, the technical and clinical success rates were 100% (4/4 patients), each in the rectum, and 75% (3/4 patients), each in the rectosigmoid colon (*p* = 0.39). The mean procedure time in the rectosigmoid colon (38.8 ± 25.4 min) was significantly longer than that in the rectum (12.3 ± 4.3 min; *p* = 0.04).

### 3.3. Surgical and Long-Term Outcomes in BTS Cases

All surgeries were performed laparoscopically without stoma creation (Table 3). The median duration from the SEMS placement to surgery was 27 (11–144) days. Regarding postoperative complications, one patient presented with anastomotic leakage (25%; treated via emergency surgery and diverting stoma creation) and another with bowel obstruction (25%). Meanwhile, none of the patients had a wound infection or an intraperitoneal abscess. The median duration of hospitalization after surgery was 16.5 (range: 8–25) days. The hospital postoperative mortality rate was 0%, and all four patients were treated with postoperative adjuvant chemotherapy.

Regarding the long-term outcomes, one of the four patients died because of pneumonia on day 344 after the SEMS placement (Appendix A). The remaining three patients survived 49–276 (mean: 182.3) days after the SEMS placement. The mean and median survival periods were 222.8 ± 126.1 and 249 (range: 49–344) days, and no recurrence occurred.

### 3.4. Long-Term Outcomes in PAL Cases

Two patients died due to the primary cancer on days 128 and 420 with stent patency. Meanwhile, two patients survived until days 62 and 99 with stent patency (Appendix A). The mean and median survival periods were 177.3 ± 164.1 and 113.5 (range: 62–420) days, and no stent obstruction occurred (Table 4). All four patients were treated with chemotherapy after the SEMS placement.

## 4. Discussion

This study showed that the newly developed proximal release-type colonic stent had reasonably high technical and clinical success rates, and it did not cause serious adverse events. None of the patients, including the four with lesions within 5 cm of the anal verge, experienced anal pain or tenesmus. The procedure time in the rectosigmoid colon was significantly longer than that in the rectum (*p* = 0.04).

Lee et al. reported that patients with rectal obstruction had a significantly lower clinical success rate [20], and Khot et al., showed that 5% of patients with a colorectal stent placement complained of severe rectal pain [21]. Moreover, Song et al., found that 62.5% of patients who underwent stenting for malignant stenosis of the colorectum within 5 cm of the anal verge complained of anal pain. This finding is attributed to technical difficulties when accurately deploying the stent with the anal side of the stent exactly aligned with the anal side of the lesion for malignant colorectal obstruction near the anal verge, and anal irritation caused by the external anal end of the stent [10]. By contrast, the newly developed proximal release-type stent can be placed easily in a precise position with the anal side aligned because it is released from the anal side under an endoscopic view. Consequently, the stent could be deployed in lesions within 5 cm of the anal verge in patients without complaints of anal pain or tenesmus, as in this study. Accordingly, this newly developed proximal release-type stent can be the primary option for lesions within 5 cm of the anal verge, and it may contribute to the expansion of the indications for colorectal stents.

In the current study, the mean procedure time of the proximal release-type stent placement in the rectosigmoid colon was significantly longer than that in the rectum. Because the proximal release-type stent is deployed with the over-the-wire method, the delivery system should be advanced over the guidewire. Further, the guidewire must be kept under appropriate tension to prevent bending during the delivery system insertion [16,22]. However, the bending of the rectosigmoid colon is extremely severe, such that it is often difficult to keep the guidewire under appropriate tension when advancing the delivery system, even though a guidewire that is thicker than that used for conventional distal release-type stent placement is used. Accordingly, the procedure time was longer in the rectosigmoid colon because it was difficult to maintain the wire tension appropriately when crossing the rectosigmoid colon that was severely bent. Therefore, for a malignant obstruction located at the oral side of the bent area in the rectosigmoid colon, conventional distal release-type stents using the through-the-scope method could be recommended for deployment.

Regarding the adverse events of the SEMS placement, one patient experienced stent migration at the rectosigmoid colon. Hence, there might be concerns regarding the use of the proximal release-type stent, as it is associated with a high risk of migration because it is deployed from the anal side. However, this stent is designed with a gentle flare toward the oral side. Therefore, the flare will prevent migration in cases of stenosis of any length [16]. Moreover, even if migration occurs, the risk of perforation is quite low due to the proximity of the stent to the anus and easy removal. Thus, this is not a major concern, and the results of this study are acceptable. Nevertheless, larger prospective studies should be conducted to completely evaluate the risk of migration.

The current research had several limitations. First, this was a small retrospective nonrandomized study with no control group. Hence, the inferences regarding causality were limited. Second, as this was an observational study, the details of the treatment strategy were determined according to each case, which might have led to bias in terms of the success rates of the SEMS placement. Third, the study was conducted at a single center in Japan using only one SEMS device, thereby limiting its generalizability. Nonetheless, this preliminary study evaluated the efficacy of the first newly developed proximal-release colonic stent.

In conclusion, this newly developed proximal release-type colonic stent was effective for lesions near the anal verge in the rectum because it facilitates placement in the right position. Hence, it can be the primary option for these types of lesions, and it may contribute to the expansion of indications for colorectal stent placement for lesions near the anal verge (<5 cm). Nevertheless, more prospective, randomized studies should be performed in the future.

## Figures and Tables

**Figure 1 jcm-11-01675-f001:**
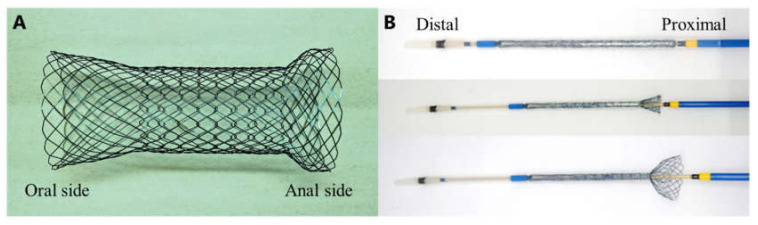
The newly developed proximal release-type colonic stent. (**A**): The diameter and length of this new stent were 22 and 70 mm, respectively. It has flares at both ends to prevent migration. (**B**): The stent was mounted on a 16 Fr delivery system and was released from the proximal side.

**Figure 2 jcm-11-01675-f002:**
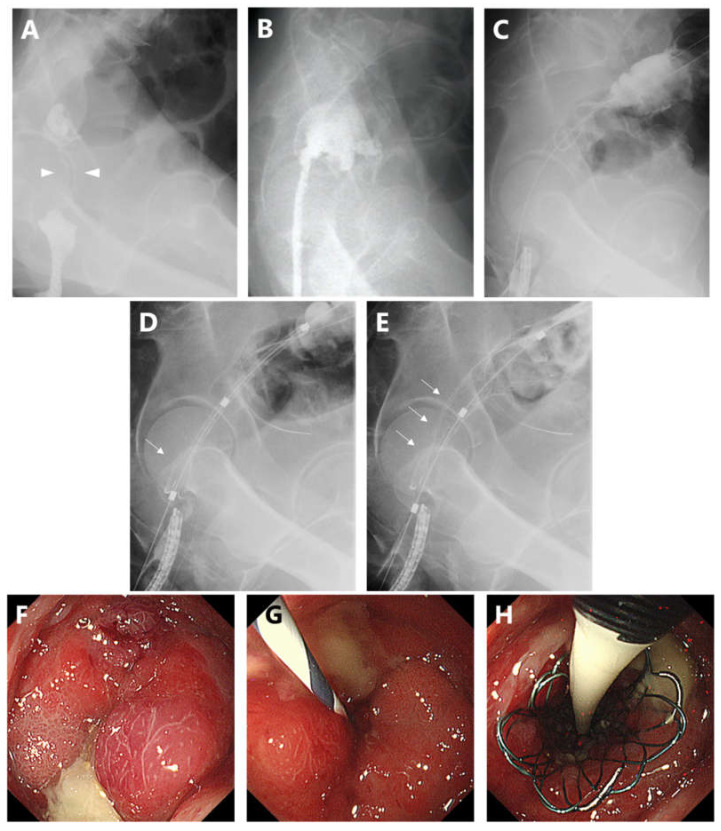
Process of stent placement. (**A**,**F**): The obstructive lesion (arrowhead) in the rectum (Rb) and the stenosis length were assessed via fluoroscopy using a contrast agent. (**B**): The nasal endoscope was advanced via the stenosis to the oral side, and a guidewire was placed across the stenosis. (**C**,**G**): The scope was removed while leaving the guidewire across the stenosis and reintroduced beside the guidewire. (**D**,**H**): The delivery system was advanced over the wire under the nasal endoscopic view, and the stent was released from the proximal side (arrow). (**E**): The deployment of the stent at a precise position (arrow) under nasal endoscopic and fluoroscopic views.

**Table 1 jcm-11-01675-t001:** Patient and tumor characteristics.

Age, years, median (range)	73.5 (64–80)
Sex, male % (n)	37.5 (3/8)
ECOG performance status score % (n)	
0–1	87.5 (7/8)
2–4	12.5 (1/8)
Etiology of colorectal obstruction % (n)	
Primary colorectal cancer	50.0 (4/8)
Anastomotic recurrence	12.5 (1/8)
Gastric cancer	25.0 (2/8)
Pancreatic cancer	12.5 (1/8)
CROSS score before SEMS placement % (n)	
0	12.5 (1/8)
1	50.0 (4/8)
2	12.5 (1/8)
3	25.0 (2/8)
4	0 (0/8)
Location of obstruction % (n)	
Rectosigmoid colon	50.0 (4/8)
Anastomosis of the sigmoid colon	12.5 (1/8)
Ra	12.5 (1/8)
Rb	25.0 (2/8)
Stricture length, cm, median (range)	4 (3–10)
Distance from the dentate line, cm, median (range)	5.5 (2–9)
Treatment intent % (n)	
Bridge to surgery	50.0 (4/8)
Palliation	50.0 (4/8)
Chemotherapy before SEMS placement % (n)	50.0 (4/8)

**Table 2 jcm-11-01675-t002:** Short-term outcomes of the newly developed proximal-release SEMS placement.

	Total	Rectal Tumors	Rectosigmoid Colon Tumors
Technical success rate % (n)	87.5 (7/8)	100 (4/4)	75.0 (3/4)
Procedure time, min, mean ± SD	25.5 ± 22.0	12.3 ± 4.3	38.8 ± 25.4 *
Clinical success rate % (n)	87.5 (7/8)	100 (4/4)	75.0 (3/4)
BTS success rate % (n)	100 (4/4)	-	100 (4/4)
Early adverse events rate % (n)	12.5 (1/8)	0 (0/4)	25.0 (1/4)
Stent migration	12.5 (1/8)	0 (0/4)	25.0 (1/4)
Perforation	0 (0/8)	0 (0/4)	0 (0/4)
Re-obstruction	0 (0/8)	0 (0/4)	0 (0/4)
Bleeding	0 (0/8)	0 (0/4)	0 (0/4)

*: *p* = 0.04.

**Table 3 jcm-11-01675-t003:** Surgical and long-term outcomes in the BTS cases.

Surgical Approach % (n)	
Open	0 (0/0)
Laparoscopy	100 (4/4)
Surgical procedures % (n)	
Low anterior resection	100 (4/4)
Without diverting stoma	100 (4/4)
Postoperative complications % (n)	
Anastomotic leakage	25.0 (1/4) *
Wound infection	0 (0/4)
Intraperitoneal abscess	0 (0/4)
Bowel obstruction	25.0 (1/4)
Overall stoma creation rate % (n)	25.0 (1/4) *
Duration from SEMS to surgery, days, median (range)	27 (11–144)
Postoperative mortality rate % (n)	0 (0/4)
Length of hospital stay, days, median (range)	16.5 (8–25)
Survival period, days, median (range)	249 (49–344)

* Emergency reoperation with a diverting stoma.

**Table 4 jcm-11-01675-t004:** Long-term outcomes in the PAL cases.

Late Adverse Events (Including Minor) % (n)	
Perforation	0 (0/4)
Stent migration	0 (0/4)
Bleeding	0 (0/4)
Duration of stent patency, days, median (range)	113.5 (62–420)
Survival period, days, median (range)	113.5 (62–420)
Mortality % (n)	50.0 (2/4)
Chemotherapy after SEMS placement % (n)	100 (4/4)

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
