# Peer review of "The Deployment of a Newly Developed Proximal Release-Type Colonic Stent Is Feasible for Malignant Colorectal Obstruction near the Anal Verge: A Single-Center Preliminary Study"

_jcm, 2022, doi:10.3390/jcm11061675_

Round 1

Reviewer 1 Report

MAJOR: 1. Could the authors be more speciffic in describing the reasons behinf the issue of why the cohort of patients was not offered side colostomy as an alternative to stenting. This should be clarified in the 2.2 subchapter

2. Can the authors be more speciffic in mentioning the exact no. of patiens undergoing emergent upfront colonoscopy exams in the obstructive settine (i.e lines 78-85, are vageu in describing patients' selection criteria)

3. Can the authors please specify why were low/mid rectum stented occlusive tumours operated upfront in the 27-days interval after stenting, and why they were not offered neoadjuvant treatment as for local disease control?

4. Can the authors extensively address the issue of potential aggravated liver metastases occuring after the stenting setting, as reported by several other simmilar cohort-studies reporting this matter for left/descending colon cancers which were bridged to surgery, via stenting?

MINOR: 

  1. Shouldn't the Abstract section be written in an IMRAD structure?
  2. Line 48 - please place "." after citation [2, 4]
  3. 3. idem line 51
  4. idem line 59. 
  5. Please consider rephrasing the "Materials and Methods" section to "Patients and Methods"
  6. Can the authors briefly describe in the extended text the acronyms BTS and PAL, as to reduce bias of understanding?
  7. line 148 - "Result" is a typo; please revise to "Results"

Author Response

Reviewer 1

Comments and Suggestions for Authors

MAJOR: 1. Could the authors be more speciffic in describing the reasons behinf the issue of why the cohort of patients was not offered side colostomy as an alternative to stenting. This should be clarified in the 2.2 subchapter

Response:

We appreciate the Reviewer’s valuable suggestion. Accordingly, we have added the following explanation in Section 2.2.

2.2 Inclusion and Exclusion Criteria

The registry included patients with colorectal obstruction within 10 cm of the anal verge caused by CRC or extracolonic cancer that required decompression for BTS or PAL. The diagnosis was based on abdominal radiography, computed tomography (CT) scan, or colonoscopy. The decision for stent placement or stoma creation was made by each patient. In a shared decision-making process, the risk of stent-related perforation, recurrence rate, overall survival and postoperative mortality, overall complication and permanent stoma rates, rate of laparoscopic one-stage surgery procedures, and technical and clinical failure rates of stenting were considered [15]. The exclusion criteria were enteral ischemia, suspected or impending perforation, intra-abdominal abscess, any contraindication to endoscopic treatment, and use of the stent for which it was not indicated. (pages 4-5, lines 89-99)

  1. Can the authors be more speciffic in mentioning the exact no. of patiens undergoing emergent upfront colonoscopy exams in the obstructive settine (i.e lines 78-85, are vageu in describing patients' selection criteria)

Response:

We appreciate the Reviewer’s inquiry and helpful suggestion. The following clarification has been added to Section 2.1.

2.1 Study Design and Participants

This single-center retrospective observational study was conducted to assess the efficacy and safety of the newly developed proximal release-type colonic stent. Among a total of 60 patients who underwent emergency endoscopy for malignant colorectal obstruction in our hospital between May 2019 and July 2021, eight patients with malignant colorectal obstructions up to 10 cm from the anal verge who underwent SEMS placement using the proximal release-type colonic stent were included in the analysis (pages 3–4, lines 70–75).

  1. Can the authors please specify why were low/mid rectum stented occlusive tumours operated upfront in the 27-days interval after stenting, and why they were not offered neoadjuvant treatment as for local disease control?

Response:

We thank the Reviewer for their comment. As per your comment, we have added the following in the revised “Study Design and “Participants” subsection.

2.1 Study Design and Participants

The time interval between stenting and surgery was dictated primarily by the general condition of the patient and the evaluation of the oral bowel, which was approximately 2 weeks. Neoadjuvant was considered only in cases with a strong tendency for invasion (page 4, lines 80–82).

  1. Can the authors extensively address the issue of potential aggravated liver metastases occuring after the stenting setting, as reported by several other simmilar cohort-studies reporting this matter for left/descending colon cancers which were bridged to surgery, via stenting?

Response:

We appreciate the Reviewer’s insightful comment as this is an important point. However, as the ESGE guideline states that the overall survival rate of stent placement is equivalent to that of emergency surgery, the issue of potentially aggravated liver metastases occurring after the stent placement was excluded from consideration in the present study.

MINOR: 

Shouldn't the Abstract section be written in an IMRAD structure?

Introduction, Materials and Methods, Results, And discussion

  1. Line 48 - please place "." after citation [2, 4]
  2. idem line 51
  3. idem line 59. 
  4. Please consider rephrasing the "Materials and Methods" section to "Patients and Methods"

Response:

We appreciate the kind recommendations of the Reviewer. We have corrected the grammatical issues pointed out by the Reviewer.

  1. Can the authors briefly describe in the extended text the acronyms BTS and PAL, as to reduce bias of understanding?

Response:

We apologize for our unclear description of the acronyms BTS and PAL. We have redrafted the Section 2.1 to establish a clearer focus as follows and to define the acronyms at their first mention.

2.1 Study Design and Participants

Treatment intent (BTS or PAL) was determined based on malignant disease stage, comorbidities, age, and patient preference. Scheduled elective resection of primary tumors within 1 year of the stent placement was classified as BTS, whereas stent placement for symptomatic relief without scheduled surgery was classified as PAL. (page 4, lines 76–80)

  1. line 148 - "Result" is a typo; please revise to "Results"

Response:

We appreciate the Reviewer’s careful observation. The grammatical error has been corrected.

Reviewer 2 Report

The manuscript entitled „The deployment of a newly developed proximal release-type colonic stent is feasible for malignant colorectal obstruction near the anal verge: A single-center preliminary study” by Kaoru Wada et al., investigates the efficacy and safety of a proximal release-type stent for malignant colorectal obstruction. Although the number of patients included in the study is small, this preliminary study is of major importance in the medical field of colorectal cancer associated with colonic obstruction upon diagnosis. In general, the technical experiments and medical procedures were well conducted in this manuscript. However, some aspects should be clarified:

  1. Abstract and the Introduction section: Please highlight the novelty of this study.

Materials and methods section:

  1. Is the proximal release-type colonic stent developed by the authors or it is purchased from a manufacturer? Please provide the detailed specifications about the stent into a subsection of the Materials and methods section.
  2. Is the Over-The-Wire method a highly used method to colonic stenting procedure? Please provide a citation for this method.
  3. Which is the contrast agent used in fluoroscopy?

Results section

  1. Certain interpreted data are repeated all over the text. Please simplify the sentences so that the information is concise and can be found only once.

Discussion section

  1. Also, the results are mentioned in this section. Please discuss your results without repeating them. It is sufficient to correlate your observations with the existing literature and to highlight the novelty and the impact of your study on the medical field.

Author Response

Reviewer 2

Comments and Suggestions for Authors

The manuscript entitled „The deployment of a newly developed proximal release-type colonic stent is feasible for malignant colorectal obstruction near the anal verge: A single-center preliminary study” by Kaoru Wada et al., investigates the efficacy and safety of a proximal release-type stent for malignant colorectal obstruction. Although the number of patients included in the study is small, this preliminary study is of major importance in the medical field of colorectal cancer associated with colonic obstruction upon diagnosis. In general, the technical experiments and medical procedures were well conducted in this manuscript. However, some aspects should be clarified:

Abstract and the Introduction section: Please highlight the novelty of this study.

Response:

We thank the Reviewer for their helpful recommendation. Accordingly, we have added the following to the revised manuscript.

  1. Introduction

To overcome this issue, a new proximal release-type colonic stent that could facilitate adjusting the position of the proximal edge was developed [16]. However, no study to date has evaluated the newly developed proximal release-type colonic stent; thus, its efficacy and safety remain unclear. Therefore, the current study aimed to investigate the efficacy and safety of this colonic stent. (page 3, lines 62–66)

Materials and methods section:

Is the proximal release-type colonic stent developed by the authors or it is purchased from a manufacturer? Please provide the detailed specifications about the stent into a subsection of the Materials and methods section.

Response:

We appreciate the Reviewer’s helpful suggestions. The proximal release-type colonic stent was designed by a group with whom we are closely acquainted. The stent was initially used only in a limited number of facilities and has recently become commercially available. We have added the detailed specifications of the stent in the Patients and Methods section as follows.

2.3 SEMS Placement Procedures

In the current study, Niti-S (Enteral Colonic Uncovered Stent; Taewoong Medical Co., Gimpo, South Korea), which is a flexible, fine mesh tubular prosthesis made of nitinol wire that contained radiopaque markers on each end and at the center, was used as the proximal release-type colonic stent. The diameter and length of the newly proximal release-type colonic stent were 22 and 70 mm, respectively, and the delivery system was 16 Fr (Fig. 1) (page 5, lines 101–106).

Is the Over-The-Wire method a highly used method to colonic stenting procedure? Please provide a citation for this method.

Response:

We appreciate the Reviewer’s insightful comment and valuable suggestion. Prior to the development of stents using the TTS method, stent placement was performed using the Over-The-Wire method. We have added the reference for the Over-The-Wire method as follows.

2.3 SEMS Placement Procedures

The diameter and length of the newly proximal release-type colonic stent were 22 and 70 mm, respectively, and the delivery system was 16 Fr (Fig. 1). The procedures were performed in a fluoroscopy room by a certified endoscopist with experience in colonic stenting using the Over-The-Wire method [17, 18] (page 5, lines 104–107).

Which is the contrast agent used in fluoroscopy?

Response:

We thank the Reviewer for their important inquiry and apologize for the oversight. We have added the name of the contrast agent in the revised manuscript as follows.

2.3 SEMS Placement Procedures

The stricture length was measured via fluoroscopy using gastrografin as the contrast agent (Fig. 2 A) (page 5, line 108).

Results section

Certain interpreted data are repeated all over the text. Please simplify the sentences so that the information is concise and can be found only once.

Response:

We appreciate the Reviewer’s helpful suggestions. We have revised the Results section as follows.

3.2 Short-Term Outcome of SEMS Placement

The technical success rate was achieved in 87.5% of patients (7/8) (Table 2). In one patient, technical failure occurred because of stent migration in the rectosigmoid colon during the procedure, and it was treated with re-SEMS placement on the same day. (page 7, lines 157–159)

In the subgroup analysis according to tumor location, the technical and clinical success rates were 100% (4/4 patients) each in the rectum and 75% (3/4 patients) each in the rectosigmoid colon (p = 0.39). Stent migration at the rectosigmoid colon occurred during the procedure in one patient. (page 8, lines 168–170)

Discussion section

Also, the results are mentioned in this section. Please discuss your results without repeating them. It is sufficient to correlate your observations with the existing literature and to highlight the novelty and the impact of your study on the medical field.

Response:

We appreciate the Reviewer’s helpful suggestion and agree with their assessment. The Discussion section has been revised accordingly as follows.

  1. Discussion

The technical and clinical success rates of the proximal release-type colonic stent placement were 87.5% each. Lee et al. reported that patients with rectal obstruction had a significantly lower clinical success rate, [20], and Khot et al. showed that 5% of patients with colorectal stent placement complained of severe rectal pain. [21]. (page 9, lines 198–200)

In the current study, the mean procedure time of the proximal released-type stent placement in the rectosigmoid colon (38.8 ± 25.4 min) was significantly longer than that in the rectum (12.3 ± 4.3 min; p = 0.04). (page 9, lines 213–214)

Round 2

Reviewer 1 Report

all MAJOR and Minor comments have been addressed